# Prevalence of Vitamin D Deficiency in Treatment-Naïve Subjects with Chronic Pulmonary Aspergillosis

**DOI:** 10.3390/jof6040202

**Published:** 2020-10-01

**Authors:** Inderpaul Singh Sehgal, Sahajal Dhooria, Kuruswamy Thurai Prasad, Valliappan Muthu, Naresh Sachdeva, Sanjay Kumar Bhadada, Ashutosh Nath Aggarwal, Mandeep Garg, Arunaloke Chakrabarti, Ritesh Agarwal

**Affiliations:** 1Department of Pulmonary Medicine, Postgraduate Institute of Medical Education and Research (PGIMER), Chandigarh 160012, India; inderpgi@outlook.com (I.S.S.); sahajal@gmail.com (S.D.); docktp@gmail.com (K.T.P.); valliappa@gmail.com (V.M.); aggarwal.ashutosh@outlook.com (A.N.A.); 2Department of Endocrinology, Postgraduate Institute of Medical Education and Research (PGIMER), Chandigarh 160012, India; naresh_pgi@hotmail.com (N.S.); bhadadask@rediffmail.com (S.K.B.); 3Department of Radiodiagnosis and Imaging, Postgraduate Institute of Medical Education and Research (PGIMER), Chandigarh 160012, India; gargmandeep@hotmail.com; 4Department of Medical Microbiology, Postgraduate Institute of Medical Education and Research (PGIMER), Chandigarh 160012, India; arunaloke@hotmail.com

**Keywords:** chronic pulmonary aspergillosis (CPA), aspergillosis, ImmunoCap, ELISA, aspergilloma

## Abstract

The association of vitamin D deficiency in the pathogenesis of invasive and allergic pulmonary aspergillosis is known. Whether vitamin D deficiency is prevalent in chronic pulmonary aspergillosis (CPA) remains unknown. We evaluated the prevalence of vitamin D deficiency in subjects with CPA. We compared the clinicoradiological features, microbiology, the immunological response, and the severity of CPA in those with or without vitamin D deficiency. We measured plasma vitamin D levels in 230 consecutive treatment-naïve subjects with CPA and 78 controls (28 with prior tuberculosis (TB); 50 healthy controls). We defined vitamin D deficiency as 25(OH)D3 level <20 ng/mL. The mean (95% confidence intervals (CI)) levels of plasma vitamin D levels were 19.5 (17.6–21.4), 18.6 (13.9–23.3), and 15.3 (12.6–17.9) ng/mL in subjects with CPA, diseased controls, and healthy controls, respectively; and the levels were not different between the groups. The prevalence of vitamin D deficiency in subjects with CPA was 65% (*n* = 150) and was also not different between healthy (70%) or diseased (61%) controls. We did not find any difference in the clinicoradiological features, microbiology, immunological response, and severity of CPA between vitamin D sufficient and deficient groups. The prevalence of vitamin D deficiency is high in subjects with CPA, albeit similar to controls. Vitamin D deficiency does not affect the disease severity in subjects with CPA.

## 1. Introduction

The fungus *Aspergillus* causes a spectrum of diseases in humans, depending on the host immunity. In those with structural lung disorders, *Aspergillus* causes chronic infection, namely chronic pulmonary aspergillosis (CPA) [1]. CPA can be categorized as simple aspergilloma (SA), chronic cavitary pulmonary aspergillosis (CCPA; presence of one or more cavities with or without a fungal ball), chronic fibrosing pulmonary aspergillosis (cavitation and fibrosis involving two or more lobes), and *Aspergillus* nodule [2,3].

CPA is believed to occur secondary to defects in lung immune responses that result in the ineffective clearing of the fungi [4,5,6]. Vitamin D, a non-essential vitamin primarily involved in bone metabolism, has also been demonstrated to have immunomodulatory effects by enhancing innate and adaptive immunity [7,8]. Vitamin D modulates the Th1, Th2, and Th17 immune responses [9], and upregulates the regulatory T cells [7,10]. Vitamin D also promotes the clearance of *Aspergillus* [8,10]. Whether vitamin D deficiency is associated with CPA or its severity is not known.

We hypothesized that subjects with CPA would have a high prevalence of vitamin D deficiency and that those with a deficiency are likely to have a severe disease. Herein, we determine the prevalence of vitamin D deficiency in subjects with CPA. We also compare the clinical, radiological, immunological, and microbiological characteristics in subjects with CPA of those with or without vitamin D deficiency.

## 2. Materials and Methods

This was a case-control study conducted between January 2019 and April 2020 in the Chest Clinic of this Institute. The Institute Ethics Committee (NK/4394/Res/512; 4th May 2018) approved the study protocol, and written informed consent was obtained from all the study participants. We have previously published some of the data [11,12].

### 2.1. Study Participants

Cases: We included consecutive treatment-naïve subjects with CPA (Appendix A). Briefly, the presence of all of the following was used to diagnose CPA: (1) respiratory symptoms for at least three months; (2) radiological features of a cavity with or without a fungal ball or pleural thickening or pericavitary infiltrates or nodules on a computed tomography (CT) of the thorax; (3) demonstration of *Aspergillus* infection by either culture or serological methods; and (4) exclusion of other pulmonary disorders with a similar presentation [2,3,11,13]. We excluded subjects with any of the following: (1) failure to provide informed consent; (2) intake of antifungal azoles for >3 weeks in the preceding six months; and (3) pregnancy. We performed a sputum or bronchoalveolar lavage fluid (BALF) smear for acid-fast bacilli, Xpert MTB/Rif, and culture for mycobacteria and bacterial pathogens to exclude other pulmonary infections.

Controls: The disease controls were subjects with previously treated pulmonary tuberculosis who had minimal symptoms or radiological abnormalities and did not fulfill the criteria for CPA. We also included 50 healthy controls. The healthy controls were the health care workers (technicians, nurses, doctors) working in the same hospital, including some of the authors of the manuscript. None of the healthy controls reported any comorbid illness and all had a normal body mass index (BMI).

Study objective: The primary outcome was to evaluate the prevalence of vitamin D deficiency in subjects with CPA. The secondary objectives were to compare the clinical, immunological, microbiological profile and the severity of CPA in those with or without vitamin D deficiency. For the study, we categorized CPA as severe if there were two or more cavities on the CT thorax or a single cavity with serum *A. fumigatus*-specific IgG >100 mg A/L.

### 2.2. Study Procedure

We performed the following investigations in the study population: serum *A. fumigatus*-specific IgE and IgG levels, serum total IgE, sputum culture for fungus, mycobacteria, and bacteria, complete blood count, C-reactive protein values, estimation of galactomannan in serum and BALF, spirometry, chest radiograph, CT of the thorax, and vitamin D levels. We evaluated the quality of life in cases and diseased controls using the St. George’s Respiratory Questionnaire [14].

Serum *A. fumigatus*-specific IgE and IgG levels: We assayed *A. fumigatus*-specific IgE and IgG using the automated fluorescent enzyme immunoassay (Phadia 100, Thermofisher Scientific, Uppsala, Sweden) [11,15]. An *A. fumigatus*-specific IgE and IgG value >0.35 kUA/L and >27 mgA/L, respectively, was considered positive [11,15].

Serum total IgE levels: We estimated the serum total IgE using the fluorescent enzyme immunoassay system (Phadia 100, Thermo Fisher Scientific, Uppsala, Sweden) [16].

Serum and BALF galactomannan: We measured the serum and BALF GM using a one-stage immune-enzymatic sandwich microplate assay (Platelia Aspergillus EIA; Bio-Rad Laboratories, Hercules, CA, USA), according to the manufacturer’s recommendations. We used the serum and BALF cut-off values of >0.6 and >1.6, respectively, to diagnose CPA [2,12,15].

Sputum and BALF fungal cultures: We used the respiratory samples treated with *N*-acetyl-l-cysteine (NALC) and sodium citrate for microscopy (calcofluor dye with potassium hydroxide) and culture (Sabouraud glucose agar medium). We identified *Aspergillus* species cultured on malt extract agar and Czapek–Dox agar by their morphological characteristics.

Vitamin D levels: We estimated the total plasma 25-hydroxyvitamin D levels by a competitive electrochemiluminescence immunoassay on a fully automated analyzer (E601, Roche Diagnostics GmbH, Mannheim, Germany). The minimum detection limit of the assay was 3 ng/mL. The inter- and intra-assay coefficient of variation was 13.1% and 6.8%, respectively. We defined vitamin D (25[OH]D3) levels ≥30 ng/mL, 21–29 ng/mL, and <20 ng/mL as normal, insufficient, and deficient, respectively [9,17].

### 2.3. Statistical Analysis

Data are presented descriptively as a mean (95% confidence interval (95% CI)) or number (percentage). We analyzed the difference between continuous and categorical variables using the Mann–Whitney U test and Fischer’s exact test, respectively. A *p*-value <0.05 was considered statistically significant.

## 3. Results

We included 230 cases with CPA and 78 controls (28 with prior TB; 50 healthy controls). The demographic profile was similar in cases and controls (Table 1). The diseased controls were minimally symptomatic and had the same duration of symptoms compared to CPA. Subjects with CPA were more likely to have a cough and recurrent hemoptysis when compared to diseased controls. The quality of life assessed by the SGRQ questionnaire was poor in subjects with CPA. The symptom and the activity domains were the most affected in subjects with CPA. The serum *A. fumigatus*-specific IgG levels were significantly higher in subjects with CPA compared to the diseased controls. There were no significant differences in the serum GM levels or other immunological parameters between CPA and diseased controls. One or more cavities on the CT thorax were universal in subjects with CPA (Table 1).

Primary outcome: The mean (95% CI) serum vitamin D level was 19.5 (17.6–21.4) ng/mL in subjects with CPA. There was no statistically significant difference in vitamin D levels among cases and controls (Table 2). The prevalence of vitamin D deficiency was 65% (*n* = 150) in subjects with CPA, like healthy (70%) and diseased (61%) controls.

Secondary outcomes: Subjects with vitamin D deficiency were significantly younger than the nondeficient group (Table 3). We did not observe any difference in the clinical parameters, pulmonary function, inflammatory markers (ESR, CRP), and quality of life in subjects with CPA with or without vitamin D deficiency. We also did not observe any difference in the immunological findings (*A. fumigatus*-specific IgE and IgG, serum total IgE), serum and BALF GM, and demonstration of *Aspergillus* in respiratory secretions in subjects with or without vitamin D deficiency. There was no difference in the underlying etiology for CPA (TB vs. non-TB), the radiological findings of the CT thorax, or the type of CPA in subjects with or without vitamin D deficiency. Importantly, vitamin D deficiency was not associated with the severity of CPA.

## 4. Discussion

We found a high prevalence of vitamin D deficiency in subjects with CPA. However, the prevalence of vitamin D deficiency was not different between diseased and healthy controls. We also found no difference in the quality of life, lung function, immunological profile, or severity of disease in subjects of CPA with or without vitamin D deficiency.

After inhalation, the *Aspergillus* conidia are effectively cleared by the body’s immune system by innate and adaptive immune responses [4,6]. Both genetic and acquired defects in the production of interleukin (IL-10), IL-12, IL-17, interferon–gamma (IFN-γ), mannose-binding lectin (MBL), toll-like receptors (TLR), and transforming growth factor–beta (TGF-β) have been implicated in CPA [18]. Vitamin D plays a vital role in immunity against infections with *Aspergillus* by enhancing the proliferation of TGF-β, producing regulatory T-cells, by an improved phagocytic activity of macrophages, and by a cathelicidin-induced enhancement in the innate immune functions of the alveolar epithelial cells [7,8,19]. Besides, vitamin D increases resistance against *Aspergillus* infection by modulating autophagy [20]. Vitamin D inhibits the formation and release of Th1 cytokines whilst inducing the Th2 cytokines [21,22]. We were also anticipating that subjects of CPA with vitamin D deficiency would demonstrate an increase in *A. fumigatus*-specific IgG levels compared to those without. However, we did not find any difference in the immunological response, severity of CPA, or quality of life in relation to vitamin D levels.

While a few studies have demonstrated a prevalence of vitamin D deficiency in allergic aspergillosis [9,23], none have evaluated the association between vitamin D deficiency and CPA. We found a high prevalence (65%) of vitamin D deficiency in subjects with CPA. The most common underlying structural lung disease in our study was the sequelae of pulmonary tuberculosis. Although vitamin D deficiency also increases the risk of tuberculosis [24,25], the prevalence of vitamin D deficiency in our study did not vary depending on the underlying etiology for CPA (TB vs. non-TB). The high prevalence of vitamin D deficiency is likely due to a high prevalence of vitamin D deficiency, in general, in the Indian population [26]. It is possible that the vitamin D cut-off for deficiency is likely to be different in the Indian population; however, the cut-off currently remains unclear. This is the reason why we chose the 20 ng/mL mark for deficiency.

What are the clinical implications of this study? We found a high prevalence of vitamin D deficiency in treatment-naïve subjects with CPA, irrespective of the underlying etiology for CPA. While vitamin D deficiency was not associated with the severity of CPA, the question of whether a replacement of vitamin D in deficient subjects improves treatment outcomes in CPA should be ascertained in future studies. Finally, our research has a few limitations, the major one being that it comprised a report from a single center. More studies from different geographic locales are needed to ascertain the prevalence of vitamin D deficiency in CPA. We also do not have follow-up details regarding treatment outcomes with antifungal agents in those with or without vitamin D deficiency. Finally, we used a convenient sample size for the study. Assuming a prevalence of vitamin D deficiency of 70% in control and 80% in CPA, with a two-sided confidence level (1-alpha) of 95% and power of 80%, we would require 600 subjects (300 cases and 300 controls).

In conclusion, there was a high prevalence of vitamin D deficiency in subjects with CPA, although it was comparable to subjects without CPA. Vitamin D deficiency was also not associated with the severity of CPA.

## Figures and Tables

**Table 1 jof-06-00202-t001:** Demographics and clinical profile of the study population.

Parameter	CPA (*n* = 230)	Diseased Controls (*n* = 28)	Healthy Controls (*n* = 50)	*p*-Value
Demographic				
Age, years	43.9 (42–45.7)	47 (40.3–53.7)	41.1 (37.2–45.1)	0.23
Male sex	117 (50.9)	19 (67.9)	29 (58)	0.19
BMI, kg/m^2^	19.6 (19–20.2)	21.2 (19.4–23)	-	0.08
Clinical findings				
Duration of symptoms, years	1.9 (1.5–2.4)	1.2 (0.5–1.8)		0.26
Cough	200 (87)	19 (67.9)		0.02
Dyspnea	52 (23.4)	4 (14.3)		0.34
Recurrent hemoptysis	156 (68.1)	8 (28.6)		<0.0001
Spirometry				
FEV1, liters	1.78 (1.54–2.03)	1.38 (1.13–1.63)	-	0.24
FVC, liters	2.24 (2.11–2.36)	2.13 (1.86–2.39)	-	0.54
FEV1/FVC ratio	72.8 (70.6–75.1)	63.7 (57.8–69.5)	-	0.006
ESR	31.4 (27.5–35.4)	33.5 (1.9–65.1)	-	0.85
Quality of life (SGRQ)				
Total score	22.5 (20.6–24.4)	16.6 (11.2–22)	-	0.05
Symptom domain	33.9 (32.1–35.7)	27.5 (21.9–33.1)	-	0.03
Activity domain	30.5 (27.7–33.3)	21.9 (13.7–30)	-	0.04
Impact domain	14.2 (11.9–16.6)	9.9 (4.6–15.2)	-	0.23
Immunological findings				
*A. fumigatus*-specific IgG, mgA/L	100.3 (92.4–108.3)	12.8 (10.3–15.4)	-	<0.0001
*A. fumigatus*-specific IgE, kUA/L	1.5 (0.9–2)	0.3 (0.1–0.8)	-	0.12
Serum Total IgE levels, IU/mL	714 (533–897)	423 (226–619)	-	0.27
Peripheral blood eosinophil count, µL	259 (230–288)	207 (140–275)	-	0.23
Serum galactomannan, ODI	0.77 (0.6–0.95)	0.39 (0.28–0.51)	-	0.13
BALF galactomannan, ODI	3.7 (3.1–4.3)	-	-	-
CT thorax findings				
Presence of cavity on CT thorax	230 (100)	12 (42.9)	-	<0.0001
Parenchymal fibrosis	179 (78.2)	23 (82.1)	-	0.81

All values are represented as a mean (95% confidence intervals) or number (percentage), unless stated; BALF: bronchoalveolar lavage fluid; BMI: body mass index; CPA: chronic pulmonary aspergillosis; CT: computed tomography; ESR erythrocyte sedimentation rate; FEV1: forced expiratory volume in one second; FVC: forced vital capacity; ODI: optical density index; SGRQ: St George’s respiratory questionnaire.

**Table 2 jof-06-00202-t002:** Vitamin D characteristics in the study population.

	CPA (230)	Diseased Controls (*n* = 28)	Healthy Controls (50)	*p*–Value
Vitamin D levels, ng/mL	19.5 (17.6–21.4)	18.6 (13.9–23.3)	15.3 (12.6–17.9)	0.13
Vitamin D status, *n* (%)				0.67
Normal	36 (15.7)	5 (17.9)	4 (8)	
Vitamin D insufficiency	44 (19.1)	6 (21.4)	11 (22)	
Vitamin D deficiency	150 (65.2)	17 (60.7)	35 (70)	

All values are represented as a mean (95% confidence intervals) or number (percentage), unless stated.

**Table 3 jof-06-00202-t003:** Demographics and clinical profile of subjects with CPA with or without vitamin D deficiency (<20 ng/mL).

Parameter	Vitamin D Deficient (*n* = 150)	Vitamin D Nondeficient (*n* = 80)	*p*-Value
Demographic			
Age, years	41 (39–43)	49 (46–52)	<0.0001
Male gender	71 (57.5)	46 (47.3)	0.09
BMI, kg/m^2^	19.5 (18.8–20.2)	19.8 (18.9–20.8)	0.61
Clinical findings			
Duration of symptoms, years	1.8 (1.4–2.3)	2.1 (1.2–3)	0.55
Cough	129 (86)	71 (88.8)	0.68
Dyspnea	36 (24.5)	16 (21.3)	0.74
Recurrent hemoptysis	101 (67.3)	55 (69.8)	0.77
Malaise	24 (16.1)	18 (22.5)	0.28
Spirometry			
FEV1, liters	1.85 (1.48–2.22)	1.66 (1.47–1.86)	0.49
FVC, liters	2.23 (2.08–2.37)	2.24 (2.01–2.48)	0.89
Inflammatory markers			
ESR	29.6 (25–34)	34.9 (27–42.8)	0.21
CRP	30 (17–43)	13 (5–21)	0.09
Quality of life (SGRQ)			
Total score	23.6 (21–26)	20.6 (17.6–23.5)	0.14
Symptom domain	33.7 (31–36)	34.2 (31–37.2)	0.79
Activity domain	32 (28–35.6)	12.6 (9–16)	0.18
Impact domain	15 (12–18)	9.9 (4.6–15.2)	0.30
Immunological findings			
*A. fumigatus*-specific IgG, mgA/L	101.1 (90.9–111.3)	98.9 (86.2–111.7)	0.80
*A. fumigatus*-specific IgE, kUA/L	1.8 (1.1–2.6)	0.9 (0.4–1.4)	0.10
Serum Total IgE levels, IU/mL	819 (568–1070)	517 (293–741)	0.12
Serum galactomannan, ODI	0.77 (0.6–0.95)	0.77 (0.42–1.1)	0.94
BALF galactomannan, ODI	3.7 (2.9–4.5)	3.6 (2.6–4.6)	0.94
Microbiology			
*Aspergillus* species	39 (23.4)	18 (19.8)	0.53
*A. fumigatus*	21	14	-
*A. flavus*	18	4	-
Underlying etiology for CPA			0.35
Tuberculosis	129 (86)	65 (81.3)	
Nontuberculosis	21 (14)	15 (18.7)	
CT thorax findings			
Number of cavities	1.8 (1.7–2)	1.8 (1.6–2)	0.70
Presence of fungal balls	93 (62)	55 (68.8)	0.39
Fungal balls	0.9 (0.8–1)	0.82 (0.7–1)	0.48
Category of CPA			0.82
Simple aspergilloma	13 (8.7)	7 (8.8)	
CCPA	112 (74.7)	63 (78.8)	
CFPA	25 (16.7)	10 (12.5)	
Severe CPA	79 (52.7)	37 (46.3)	0.41

All values are represented as a mean (95% confidence intervals) or number (percentage), unless stated; BALF: bronchoalveolar lavage fluid; BMI: body mass index; CCPA: chronic cavitary pulmonary aspergillosis; CFPA: chronic fibrosing pulmonary aspergillosis; CPA: chronic pulmonary aspergillosis; CRP: C–reactive protein; CT: computed tomography; ESR: erythrocyte sedimentation rate; FEV1: forced expiratory volume in one second; FVC: forced vital capacity; ODI: optical density index; SGRQ: St George’s respiratory questionnaire.

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
