# Peer review of "Prevalence of Vitamin D Deficiency in Treatment-Naïve Subjects with Chronic Pulmonary Aspergillosis"

_jof, 2020, doi:10.3390/jof6040202_

Round 1

Reviewer 1 Report

A well-written piece. Clinically, vitamin D deficiency is easily treatable and therefore these results are relevant, regardless of the clear limitations of the study and intrinsic to its study design. I hope to see clinical follow up in the future. 

Study participants:

Did bacterial pathogen culture include specific culture for Actinomyces?

For valid comparison, more descriptives of the healthy controls are needed. To identify bias; how were these healthy controls selected? Is the medical history known? How was the material acquired? Same hospital, same time period? Why this number of controls, did the authors perform a power calculation for control sample size, or how was this ascertained? is BMI known? etc.

Also, how many people were excluded for analysis? If extensive, include a consort diagram.

Discussion:

With IgG and IgE being important diagnostics, I would find it interesting to read on the authors interpretation of the immunological findings in the discussion.

Author Response

Response to reviewer’s comments:

We thank the reviewers for their comments. The manuscript has benefitted immensely by these comments

Reviewer 1

Comment: A well-written piece. Clinically, vitamin D deficiency is easily treatable and therefore these results are relevant, regardless of the clear limitations of the study and intrinsic to its study design. I hope to see clinical follow up in the future. 

Response: Thank you

Study participants

Comment: Did bacterial pathogen culture include specific culture for Actinomyces?

Response: We did not routinely use special cultures for actinomyces or nocardia. The special culture for actinomycetes or nocardia was only used if microscopy identified filamentous bacteria on microscopy or if there was a clinical suspicion based on radiology (nodules, chest wall involvement and others).

Comment: For valid comparison, more descriptives of the healthy controls are needed. To identify bias; how were these healthy controls selected? Is the medical history known? How was the material acquired? Same hospital, same time period? Why this number of controls, did the authors perform a power calculation for control sample size, or how was this ascertained? is BMI known? etc.

Response: The healthy controls were the health care workers (technicians, nurse, doctors) working in the same hospital and included some of the authors of the manuscript. None of the healthy controls reported any comorbid illness and had normal BMI. We used a convenient sample size. We have included this as a study limitation [lines 69-71; 184-187]

Comment: Also, how many people were excluded for analysis? If extensive, include a consort diagram.

Response: We included all consecutive subjects with CPA and did not exclude any participants from the analysis

Discussion:

Comment: With IgG and IgE being important diagnostics, I would find it interesting to read on the authors interpretation of the immunological findings in the discussion.

Response:  We have included a paragraph of immunological findings with vitamin D deficiency [lines 162-166]

Reviewer 2 Report

It is a well-documented study wherein the authors studied the correlation, if there any, between chronic pulmonary aspergillosis (CPA) and vitamin D. Comparing vitamin D level of control individuals with those having CPA, my question is that – what is the average vitamin D level in control population of India? If it is already lower than 20 ng/mL in the control group of the Indian population, then it will be better to state that there is no correlation between CPA and plasma vitamin D level, as ‘deficiency’ may not a right word.

Author Response

Response to reviewer’s comments:

We thank the reviewers for their comments. The manuscript has benefitted immensely by these comments

Reviewer 2

Comment: It is a well-documented study wherein the authors studied the correlation, if there any, between chronic pulmonary aspergillosis (CPA) and vitamin D. Comparing vitamin D level of control individuals with those having CPA, my question is that – what is the average vitamin D level in control population of India? If it is already lower than 20 ng/mL in the control group of the Indian population, then it will be better to state that there is no correlation between CPA and plasma vitamin D level, as ‘deficiency’ may not a right word.

Response: The average vitamin D levels in the healthy controls from India ranges between 3.2 and 52.9 ng/mL, with higher levels being reported from southern part of the country. The vitamin D level has been reported between 26 and 31 ng/mL in two previous studies from Chandigarh. While we agree that the vitamin D cutoff for deficiency is likely to be different in the Indian population, however currently the cutoff remains unclear. This is the reason why we have chosen the 20 ng/ml value. We have included this point in the discussion [lines 173-176]